# RPE-Directed Gene Therapy Improves Mitochondrial Function in Murine Dry AMD Models

**DOI:** 10.3390/ijms24043847

**Published:** 2023-02-14

**Authors:** Sophia Millington-Ward, Naomi Chadderton, Laura K. Finnegan, Iris J. M. Post, Matthew Carrigan, Rachel Nixon, Marian M. Humphries, Pete Humphries, Paul F. Kenna, Arpad Palfi, G. Jane Farrar

**Affiliations:** 1The School of Genetics and Microbiology, Smurfit Institute of Genetics, Trinity College Dublin, D02 VF25 Dublin, Ireland; 2The Research Foundation, Royal Victoria Eye and Ear Hospital, D02 XK51 Dublin, Ireland

**Keywords:** age-related macular degeneration, retina, gene therapy, AAV, RPE, *VMD2*, promoter, mitochondria, BEST1

## Abstract

Age-related macular degeneration (AMD) is the most common cause of blindness in the aged population. However, to date there is no effective treatment for the dry form of the disease, representing 85–90% of cases. AMD is an immensely complex disease which affects, amongst others, both retinal pigment epithelium (RPE) and photoreceptor cells and leads to the progressive loss of central vision. Mitochondrial dysfunction in both RPE and photoreceptor cells is emerging as a key player in the disease. There are indications that during disease progression, the RPE is first impaired and RPE dysfunction in turn leads to subsequent photoreceptor cell degeneration; however, the exact sequence of events has not as yet been fully determined. We recently showed that AAV delivery of an optimised NADH-ubiquinone oxidoreductase (NDI1) gene, a nuclear-encoded complex 1 equivalent from *S. cerevisiae*, expressed from a general promoter, provided robust benefit in a variety of murine and cellular models of dry AMD; this was the first study employing a gene therapy to directly boost mitochondrial function, providing functional benefit in vivo. However, use of a restricted RPE-specific promoter to drive expression of the gene therapy enables exploration of the optimal target retinal cell type for dry AMD therapies. Furthermore, such restricted transgene expression could reduce potential off-target effects, possibly improving the safety profile of the therapy. Therefore, in the current study, we interrogate whether expression of the gene therapy from the RPE-specific promoter, Vitelliform macular dystrophy 2 (*VMD2*), might be sufficient to rescue dry AMD models.

## 1. Introduction

Age-related macular degeneration (AMD) is a devastating and progressive degenerative disorder of the macula leading to loss of central vision. It is the leading cause of blindness in the elderly in the developed world, affecting ~10% of people over 65 years of age [1]. AMD is a multifactorial condition with genetic and environmental factors known to contribute to the disease, with age being the greatest risk factor. Twin studies estimate the heritability of AMD at between 46–71% [2]. AMD is generally divisible into 2 distinct forms: neovascular (wet) and non-exudative (dry), accounting for 10–15% and 85–90% of cases, respectively [3].

In early dry AMD, drusen are formed between the Bruch’s membrane (BM) and the basal lamina of the retinal pigment epithelium (RPE). In the late stage, the condition may progress to geographic atrophy (GA), characterised by atrophy of the photoreceptors, RPE and choriocapillaris in the macula. GA typically initiates in the perifoveal macula, but subsequently may expand into the cone-dominated fovea, with associated loss of central vision [4].

Another hallmark feature of AMD is chronic inflammation. Indeed, an analysis of drusen from post-mortem eyes of AMD patients demonstrated the presence of components from the complement system [5,6]. Variants in multiple complement factor genes have been linked to increased complement system activation in AMD [7]. While not all mechanisms are understood, it is known that in AMD the RPE and choriocapillaris provide less oxygen and glucose to photoreceptor cells, contributing to their death [8]. The RPE functions as an outer blood—retinal barrier, is a scavenger of photoreceptor outer segments and is important in the maintenance of retinal homeostasis, supporting photoreceptors. Electron microscopy of RPE from donor AMD patients revealed fewer and smaller mitochondria, suggesting defects in mitochondrial biogenesis and fusion, as well as abnormal mitochondrial membranes and disrupted cristae [9,10]. Indeed, RPE from AMD patients displayed reductions in autophagic flux [11], mitochondrial respiration and ATP production [11,12], and elevated levels of ubiquitin-binding protein p62 [13]. Proteomics on RPE from AMD patients showed elevated levels of mitofilin, which aids in maintaining cristae integrity, and increases in several mitochondrial chaperones, which assist with the import and folding of nuclear-encoded proteins into the mitochondria [14,15,16]. These findings strongly suggest the pivotal role that mitochondrial dysfunction in RPE plays in AMD disease progression. Notably, the RPE is exposed to high levels of reactive oxygen species (ROS) due to the significant metabolic demands of the retina [17,18]. Evidence is accumulating that elevated ROS, as well as mitochondrial dysfunction in the RPE, leads to organelle dysfunction, impaired mitophagy, DNA mutations, protein and lipid damage, and reduced respiration and/or generation of toxic lipid-derived adducts, and contributes to RPE and photoreceptor cell death in dry AMD [19,20,21,22,23,24,25,26,27,28,29].

The field of AMD has been greatly hampered by the lack of suitable disease models. Aged rodent models simulate aspects of dry AMD; however, none fully mirror AMD [29]. Notably, non-primate retinas have no macula, which limits their morphological commonality, and therefore their utility as model systems. The *Cfh^−/−^* mouse [30,31] has been used for AMD studies since the discovery that *CFH* polymorphisms are frequently associated with AMD [7,32,33,34,35]. Aged *Cfh^−/−^* mice have been shown to display impaired visual function, thinning of the outer nuclear layer and abnormal BM, while in other studies thickening of BM and basal laminar deposits (BlamDs) were reported [31,36,37]. BM deposits in *Cfh^−/−^* mice may be due to competition between CFH and lipoproteins for binding to BM, leading to lipoprotein accumulation [37]. We recently reported that aged *Cfh^−/−^* mice displayed abnormal cone morphology, with disorganised outer segment membranes. Additionally, cone inner segments contained abnormal mitochondria that were irregular in shape and generally smaller than mitochondria in adjacent rod inner segments [38]. Others have shown that *Cfh^−/−^* mice have decreased plasma C3 levels [31]. Interestingly, when *CFH* was silenced in vitro in an RPE cell line, increased inflammation, metabolic impairment and vulnerability towards oxidative stress were observed [39]. More recently, the same group silenced *CFH* in an RPE cell line, which was co-cultured with either primary retinal explants from the porcine visual streak, an area akin to the human macula, or with a human macula. The co-cultured retinal cells exhibited signs of degeneration (with rod cells seemingly being the first to suffer) and changes in mitochondria and lipid composition [40].

In contrast to the *Cfh^−/−^* model of dry AMD, chemically-induced models of AMD such as the well-established sodium iodate (NaIO_3_)-induced model [41,42,43,44] do not require aging. NaIO_3_, a strong oxidising agent, causes catastrophic damage to the RPE, leading to photoreceptor loss and reduced retinal function, including reduced electroretinogram (ERG) amplitudes [38,43,45,46,47,48].

In AMD both the RPE and photoreceptors have been shown to display mitochondrial complex 1 deficiency [16,26,46,47,48,49,50,51]. Complex 1, a component of the mitochondrial electron transport chain (ETC), is a 45-subunit complex, 7 of which are mitochondrially encoded [51]. We have recently demonstrated significant therapeutic benefit in a variety of dry AMD models with a gene therapy, ophNdi1 [38]. OphNdi1 is a modified gene from *S. cerevisiae*, NDI1, which has been optimised to express more efficiently in mammalian cells (patent no. 10220102). Encoded by a single nuclear gene, NDI1 performs a similar function to mammalian mitochondrial complex 1. OphNdi1, delivered subretinally via recombinant adeno-associated virus 2/8 (AAV2/8), was shown to provide significant and robust benefit in the *Cfh*^−/−^ and NaIO_3_-induced murine models and two cellular models of dry AMD [38]. The study was the first to show functional benefit in vivo in murine models of dry AMD, using a gene therapy which directly targets mitochondrial dysfunction. In the study, AAV2/8-ophNdi1 was driven from generic CMV and CAG promoters so that ophNdi1 was expressed in both RPE and photoreceptor cells. In contrast, in the current study, we evaluated whether targeted treatment of the RPE with ophNdi1 was sufficient to provide benefit to dry AMD models, since it is thought that AMD may initiate in the RPE prior to photoreceptor cells [52]. In addition, restricting ophNdi1 expression to the RPE may reduce potential off-target effects in other cell types; thereby, in principle, improving the potential safety profile of the therapy. Hence, in this study we replaced the generic promoter (*CMV* or *CAG*) with a vitelliform macular dystrophy (*VMD2*) promoter, which, in the context of the eye, is known to restrict gene expression to the RPE [53], creating VMD2-ophNdi1. VMD2-ophNdi1 was expressed from recombinant AAV2/8 (AAV2/8-VMD2-ophNdi1; Figure 1) and its therapeutic potential was investigated in the *Cfh^−/−^* genetic and NaIO_3_-induced murine models of AMD. Primary porcine RPE (pRPE) models of dry AMD were also utilised; these were either insulted with NaIO_3_ [38] or loaded with retinylidene-N-retinylethanolamine (A2E) and insulted with blue light [47]. A2E is a toxic bisretinoid byproduct of the visual cycle and a major component of drusen, which is known to lead to RPE dysfunction and cell death in vitro and in vivo [47,54,55,56].

## 2. Results

In order to estimate the dose of AAV2/8-VMD2-ophNdi1 (Figure 1A) that might be effective in murine RPE, we generated an AAV2/8 vector expressing *VMD2* promoter-driven enhanced green fluorescent protein (*EGFP*) gene (AAV2/8-VMD2-EGFP; Figure 1B) which could be compared in vivo to a similar *CMV*-driven *EGFP* vector (AAV2/8-CMV-EGFP). Two-month-old 129 S2/SvHsd mice were subretinally injected with 3 × 10^9^ vg of either AAV2/8-VMD2-EGFP or AAV2/8-CMV-EGFP. Relative EGFP protein expressions from the two promoters were determined in mouse retinas four weeks post-injection. EGFP expression from the *VMD2* promoter was restricted to the RPE, whereas the *CMV* promoter efficiently drove EGFP expression in the RPE and the outer nuclear layer containing the photoreceptors (Figure 1C–E). Levels of EGFP fluorescence in the RPE were evaluated using fluorescent microscopy and were estimated to be 6.5-fold higher from the *CMV* than from the *VMD2* promoter. Therefore, to account for this differential in expression, doses of AAV2/8-VMD2-ophNdi1 used in the current study, in both in vitro and in vivo models, were 6.5-fold higher than doses of the non-specific *CMV* promoter [38].

Two-month-old 129 S2/SvHsd mice were subretinally injected with 3 × 10^9^ vg of AAV2/8-CMV-EGFP and AAV2/8-VMD2-EGFP (Figure 1A). Four weeks post-injection, eyes were enucleated, fixed in 4% PFA, cryosectioned, and processed for and analysed by fluorescent microscopy. Green and blue represent EGFP and DAPI (nuclear counterstain) fluorescence, respectively. Panels C and D were displayed using the full intensity range of EGFP fluorescence. Panel E corresponds to the same microscopy image as panel D; however, in panel E, the displayed EGFP fluorescence intensity range was focused on the lower intensity values, which enabled the visualisation of lower EGFP levels. EGFP fluorescence intensity levels were evaluated in the RPE for both promoters and levels were found to be ~6.5-fold lower from *VMD2* compared to *CMV*.

### 2.1. Rescue in pRPE Cell Models

To determine whether the *VMD2* promoter-driven ophNdi1 therapy could rescue cell models of dry AMD, insulted primary pRPE cells were utilised. Primary pRPE cells isolated from *n* = 3 adult pigs were transduced with AAV2/8-VMD2-ophNdi1 (MOI = 3.4 × 10^6^), insulted with 6 mM NaIO_3_ and compared to control cells. Cells were fixed 24 h post-insult and analysed with immunocytochemistry for 8-OHdG (oxidative stress marker), CPN60 (mitochondrial marker), phalloidin (selective for F-actin) and Hoechst (nuclear stain). NaIO_3_-treated cells exhibited high levels of oxidative stress and absence of actin filaments compared to non-insulted cells, indicating severe stress and reduced viability. In contrast, insulted cells transduced with AAV2/8-VMD2-ophNdi1 appeared more similar to non-insulted controls. Furthermore, mitochondrial staining was more intense and punctuated in insulted cells that had not received therapy, likely indicating mitochondrial dysfunction. Mitochondria of AAV2/8-VMD2-ophNdi1-treated cells insulted with NaIO_3_ were more similar to non-insulted control cells, although staining was still somewhat elevated (Figure 2A–O).

The bioenergetic response to AAV2/8-VMD2-ophNdi1 treatment was also profiled in pRPE cells, isolated from *n* = 3 adult pigs. Cells were seeded (1.375 × 10^4^ cells/well) in XFe96 Seahorse plates and transduced with AAV2/8-VMD2-ophNdi1 (MOI = 3.4 × 10^6^). After 28 h, cells were insulted with 30 μM A2E and insulted with blue light for 3 h. A mitochondrial stress test was subsequently performed (Figure 2P–S, Appendix A). The A2E/blue light insult reduced basal and maximal oxygen consumption rates (OCRs), spare respiratory capacity (SRC, the difference between maximal and basal OCRs; *p* < 0.01) and ATP production in primary pRPE cells. Notably, treatment of A2E-insulted cells with AAV2/8-VMD2-ophNdi1 increased basal and maximal OCRs, and ATP production (ANOVA and post-hoc Tukey; Figure 2P–S, Appendix A).

### 2.2. Rescue of the Cfh^−/−^ Mouse

AAV2/8-VMD2-ophNdi1 clearly rescued morphological and bioenergetic damage in the A2E/blue light and NaIO_3_-induced pRPE models of dry AMD (Figure 2). To interrogate whether rescue could also be achieved in vivo with *VMD2*-ophNdi1, two-month-old *Cfh*^−/−^ mice were subretinally injected with 4.5 × 10^8^ vg AAV2/8-VMD2-ophNdi1 (*n* = 13) in one eye. The contralateral control eye received an equal volume (3 μL) of PBS. At 8 months of age electroretinography was performed on injected mice. Notably, eyes treated with AAV2/8-VMD2-ophNdi1 showed significantly greater Rod b (53.0 ± 26.0 μV versus 97.7 ± 36.2 μV, Figure 3A), Max b (111.4 ± 49.0 μV versus 178.7 ± 68.3 μV, Figure 3B) and single flash cone (SFC) b (19.8 ± 8.2 μV versus 24.4 ± 9.2 μV, Figure 3C) responses than contralateral eyes, indicating benefit in both rod and cone photoreceptor cells, in treated versus control eyes (paired *t*-test; Figure 3A–C). At 7 months, ROS levels in live cells from treated and untreated dissociated *Cfh^−/−^* retinas were compared (*n* = 10 mice) using a CellRox^TM^ assay. This assay measures ROS predominantly in photoreceptors cells, as the RPE is not isolated with the neural retina during sample collection for this assay. Notably, ROS were significantly reduced in treated samples compared to control samples (91.4 ± 8.9% versus 100 ± 8.4% respectively; *p* < 0.05, paired *t*-test, Figure 3D). In addition, at 8 months, mice were sacrificed, their eyes fixed and retinal histology performed. Cone cells in retinal sections were stained with Arr3 immunohistochemistry and quantified (Figure 3E,F). Outer nuclear layer (ONL) thickness was measured and was 41.87 ± 15.67 μm and 49.64 ± 7.762 μm (*n* = 6; *p* = 0.0795; paired *t*-test; Figure 3G) in control and treated retinas, respectively. However, this difference was a trend. Cone numbers were also determined in retinas and were 150.3 ± 52.6 cones/mm and 169.8 ± 11.6 cones/mm (*n* = 6; *p* = 0.46; paired *t*-test; Figure 3H) in control and transduced samples, respectively; however, this minor increase in cone numbers in treated retinas was not significant.

### 2.3. Rescue of the NaIO_3_-Induced Mouse

Notably, AAV2/8-VMD2-ophNdi1 provided clear functional benefit in aged *Cfh*^−/−^ mice and reduced ROS levels. However, histological benefit could not be demonstrated. Given the multifactorial nature of AMD, involving genetic and environmental factors, and limitations of available models, it was valuable to assess AAV2/8-VMD2-ophNdi1 in an additional murine model, the NaIO_3_ model. Four-month-old 129 S2/SvHsd mice received 4.5 × 10^8^ vg AAV2/8-VMD2-ophNdi1 subretinally in one eye and 1.0 × 10^8^ vg of AAV2/2-CAG-EGFP, as a marker, in both eyes. Two months post-injection, mice received 22 mg/kg NaIO_3_ via tail vein to induce acute and severe oxidative damage in the RPE, mimicking aspects of dry AMD. ERG and OKR analyses were performed one week and four weeks post-NaIO_3_ insult, respectively. ERG readings in NaIO_3_-insulted mice were reduced to the extent that only Max b responses remained reliably recordable. Max b readings in treated versus untreated eyes were 185.3 ± 154.9 μV versus 121.8 ± 44.9 μV, respectively. However, this difference was only a trend (*n* = 7, Figure 4A). AAV2/8-VMD2-ophNdi1-treated eyes displayed significantly better OKR tracking responses than control eyes (0.234 ± 0.0652 μV cyc/deg versus 0.171 ± 0.0492 cyc/deg, respectively; *n* = 7, *p* < 0.005, paired *t*-test, Figure 4B).

An additional group of mice was subretinally injected with 4.5 × 10^8^ vg AAV2/8-VMD2-ophNdi1 and 1.0 × 10^8^ vg AAV2/2-CAG-EGFP in one eye and PBS containing 1.0 × 10^8^ vg AAV2/2-CAG-EGFP in contralateral eyes and was insulted with 50 mg/kg NaIO_3_ delivered via tail vein at two months post-treatment. Retinas were fixed 7 days post-insult, cryosectioned and stained for Arr3 (cone-specific marker; Figure 4C). Cones were quantified; however, numbers were similar in treated and untreated eyes: 124.8 ± 11.3 cones/mm and 129.1 ± 21.1 cones/mm, respectively (*n* = 7, *p* = 0.56, paired *t*-test). OKR responses suggest that ophNdi1 provided some benefit. However, treatment was not sufficient to rescue ERG or histology significantly from this severe NaIO_3_ insult.

## 3. Discussion

We have previously shown that subretinally injected AAV-delivered ophNdi1 provided robust functional benefit and increased mitochondrial function in the *Cfh*^−/−^ and NaIO_3_-induced mouse models of dry AMD. Additionally, increased cellular bioenergetics and reduced cellular stress markers were found [38]. In the prior study, ophNdi1 expressed from a recombinant AAV2/8 vector was driven from a ubiquitous *CMV* or *CAG* promoter and therefore was expressed in multiple retinal cells, including RPE, rod and cone photoreceptor cells; the cells lost in advanced dry AMD. In the current study, a higher absolute dose of AAV2/8-VMD2-ophNdi1 (4.5 × 10^8^ vg) was utilised to match the effective dose range of AAV2/8-CMV-ophNdi1 (1 × 10^7^ vg and 7.5 × 10^7^ vg) and achieve comparable levels of expression of ophNdi1 from the RPE-specific *VMD2* promoter, which was estimated to express ~6.5-fold less than the *CMV* promoter in RPE cells (Figure 1C–E).

There is strong evidence that AMD may initiate in the RPE, causing subsequent dysfunction in photoreceptor cells and ultimately, in GA, cell death in RPE and photoreceptor cells. However, this order of events is by no means definitively established. Differentially expressed genes have been identified in both RPE and cells of the neural retina and in both peripheral and macular regions of post-mortem AMD patient eyes [57]. In addition, rod photoreceptor cells also show early functional and histological signs of degeneration, as early signs of AMD include parafoveal scotomas and scotopic sensitivity; the parafoveal area is dense in rod photoreceptor cells [58]. Physiological abnormalities in cones in early dry AMD have also been widely reported and are indicative markers of the severity of dry AMD [59]. The morphological changes in cones in early dry AMD are, however, more subtle and include abnormal immunoreactivity to cone opsin, in combination with swelling of and altered immunoreactivity in the cone distal axon [60]. Thus, based on these features, many researchers believe that photoreceptor cell dysfunction may possibly occur in parallel with, or even prior to, dysfunction in the RPE/Bruch’s membrane complex [61]. Hence, the exact sequence of disease progression remains somewhat obscure.

Experimental therapies for AMD have focused on preservation of photoreceptor cells, RPE or both [62,63,64,65,66,67]. However, clearly, in terms of optimising efficacy and safety and reducing possible off-target effects, there may be a therapeutic advantage to restricting expression of a gene therapy to cell types underlying the condition, with the aim of preventing the disease from progressing to other retinal cells. The aim of the current study was to determine whether expression of ophNdi1 solely in RPE is sufficient to rescue a variety of AMD cell and murine models. Primary RPE cells were insulted with either NaIO_3_ or A2E/blue light. AAV2/8-VMD2-ophNdi1 was shown to rescue these cellular models using assays for ROS, cell viability, mitochondrial morphology and mitochondrial function (OXPHOS), in a similar fashion to *CAG*-driven ophNdi1 (Figure 2), indicating that the *VMD2* promoter is functional and efficient in primary RPE cells.

In the *Cfh^−/−^* mice, ERG readings were significantly improved, and ROS levels reduced by, AAV2/8-VMD2-ophNdi1 treatment, as had also been seen in a previous study utilising AAV2/8-CAG-ophNdi1 in this model [38]. However, in contrast to AAV2/8-CAG-ophNdi1, AAV2/8-VMD2-ophNdi1 did not rescue cone numbers in the model, whereas the ONL showed a trend towards being thicker in treated eyes (Figure 3). Additionally, in the very acute and severe NaIO_3_-induced murine model of dry AMD, only OKR benefit could be demonstrated with AAV2/8-VMD2-ophNdi1 (Figure 4), whereas AAV2/8-CAG-ophNdi1 had previously provided OKR, ERG and histological benefit [38]. The data in these dry AMD models highlight the involvement of both the RPE and photoreceptors and suggest that expression of the ophNdi1 gene in both RPE and photoreceptors may be preferable to that in RPE alone.

Whether a higher dose of AAV2/8-VMD2-ophNdi1 would have provided similar benefit to *CMV*-driven ophNdi1 in the NaIO_3_-induced mice was not investigated in the current study, as the dose used with the VMD2-driven therapy was already 6-fold higher than the highest dose of *CMV*-driven ophNdi1 used previously. Note that there is an increasing focus in the field of virally-delivered gene therapy on lowering dose requirements of AAV gene therapies, thereby reducing the risk of immune responses.

It is notable that an allied experimental approach has been explored previously for ABCA4-linked Stargardt disease (STGD1) in genetically modified mice. In the study, the ABCA4 gene was expressed in the RPE, but not photoreceptors, providing partial rescue of the disease and suggesting a role for both the RPE and photoreceptors in the pathogenesis of STGD1 [68]. Both that study and our own highlight the value of differential promoter constructs to explore the relative contributions of different cell types to the pathogenesis of disease and the optimal target cell population in therapeutic interventions.

In summary, while AAV2/8-VMD2-ophNdi1, which only expresses in RPE, did provide some benefit in two murine models of dry AMD, AAV2/8-CMV-ophNdi1, which expresses in RPE and rods and cones, amongst other cell types, provided more robust and consistent benefit using a variety of functional and histological assays. It remains unclear which cells are affected first in dry AMD, RPE or photoreceptor cells. However, ubiquitous and RPE-specific promoter-driven gene therapies can be used to interrogate the contribution of different cell types to disease pathogenesis and the optimal target cell population for a therapy. In the current study, the data from two murine models suggests that using a general promoter to drive expression of ophNdi1 and boosting mitochondrial function in both RPE and photoreceptor cells is more beneficial than targeting the RPE alone.

## 4. Materials and Methods

### 4.1. Study Design

An optimised complex I equivalent gene, ophNdi1 driven from an RPE-specific promoter, *VMD2*, was delivered via recombinant AAV2/8 to models of dry AMD: the *Cfh^−/−^* mouse, NaIO_3_-induced mouse and primary pRPE cells insulted with NaIO_3_ or A2E/blue light. Functional benefit was determined using physiologic readouts, ERG and OKR. Histological analysis and cellular assays included mitochondrial function, ROS and morphological readouts.

### 4.2. Plasmid Construct, AAV Production and Analysis of Relative Expression Levels

A 547 bp region of the *VMD2* promoter was PCR-amplified from DNA using the following primer pair and was cloned upstream of ophNdi1 [38] using SacI and EcoR1. *VMD2* F primer −585: 5′ CATGAGAGCTCAATTCTGTCATTTTACTAGGGT 3′ and *VMD2* R primer +38: 5′ CATGAGAATTC GGTCTGGCGACTAGGCTGGT 3′ [53]. The same promoter region was PCR-amplified with the following F primer and the same R primer as above and the product cloned upstream of an *EGFP* gene. *VMD2* F −585 NotI: 5′ CATGAGCGGCCGCAATTCTGTCATTTTACTAGGGT 3′. *VMD2*-ophNdi1 and *VMD2*-EGFP were packaged into recombinant AAV2/8 (AAV2/8-VMD2-ophNdi1 and AAV2/8-VMD2-EGFP), and their viral titres determined as described [38]. Expression of EGFP from the *CMV* [69] and *VMD2* was compared in murine retinas by subretinally injecting adult 129 S2/SvHsd mice with 3 × 10^9^ vg of either AAV2/8-VMD2-EGFP (*n* = 3) or AAV2/8-CMV-EGFP (*n* = 2). Mice were sacrificed 1 month post-injection, eyes fixed in 4% PFA, cryosectioned and analysed by fluorescent microscopy as described [69]; mean EGFP fluorescence intensity levels were determined in the RPE.

### 4.3. Cellular Models

Primary pRPE cells were isolated from mature pig eyes (*n* = 3 pigs) and maintained in culture [70]. Cells numbering 5.0 × 10^4^ were seeded into XFe96 Seahorse plates (*n* = 3; Agilent, Santa Clara, CA, USA). A minimum of 5 wells were transduced with AAV2/8-VMD2-ophNdi1 (MOI = 3.4 × 10^6^) 24 h later. At 28 h post-transduction, transduced and untransduced cells (>15 wells per group) were insulted with 30 μM A2E (Orga-Link, Magny-les-Hameaux, France) and maintained for 3 h under blue light of ~1 mW/cm^2^ (80–90 Lux) at 430 nm. Cells then underwent a mitochondrial stress test and readings were normalised to total protein as described [38]. Mitochondrial stress tests on RPE from 3 pigs (RPE1-3) were performed on 3 separate occasions. Cells numbering 7.5 × 10^4^ pRPE (from *n* = 3 pigs) were seeded onto 8-well imaging slides (Miltenyi Biotec, Bergisch Gladbach, Germany). At 5 h post-seeding, cells were transduced with AAV2/8-VMD2-ophNdi1 (MOI = 3.4 × 10^6^). At 28 h post-transduction, cells were insulted with 6 mM NaIO_3_ and at 24 h post-insult cells were fixed in 4% paraformaldehyde in PBS at RT for 20 min. Cells were stained and analysed as described [38].

### 4.4. Subretinal Injections, Electroretinography and Ros Assay

All animal work was performed in accordance with the European Union (Protection of Animals used for Scientific Purposes) Regulations 2012 (S.I. no. 543 of 2012) and the Association for Research in Vision and Ophthalmology (ARVO) statement for the use of animals, and approved by the animal research ethics committee in Trinity College Dublin (Ref. no. 140514/240320). C57BL/6J, *Cfh^−/−^* on a pure C57BL/6J background and 129 S2/SvHsd mice (Harlan Laboratories, Blackthorn, UK.) were maintained under specific pathogen-free conditions. Injections were performed on two-month-old mice as described, except that anaesthesia comprised of ketamine and medetomidine (57 mg/kg and 0.5 mg body weight, respectively) and, following injection, an anaesthetic-reversing agent (Atipamezole Hydrochloride, 1.33 mg/kg body weight) were delivered by intraperitoneal injection [71]. An amount of 4.5 × 10^8^ vg of AAV2/8-VMD2-ophNdi1 was injected into *Cfh*^−/−^ mice, while contralateral eyes received the same volume (3 µL) of PBS. At 8 months, ERG responses from treated eyes were compared to fellow eyes (*n* = 13 mice; paired *t*-tests). Mice were analysed histologically at 8 months of age as described (*n* = 6) [38]. At 7 months post-injection, mice were sacrificed, retinal cells were dissociated and a CellRox^TM^ Green Reagent (Invitrogen, Waltham, MA, USA) ROS assay was performed using a flow cytometry assay, as described [38]. Median levels of CellRox^TM^, representing relative ROS levels, were recorded. Paired *t*-tests of the means were performed to compare medians of treated versus untreated eyes of *Cfh^−/−^* mice (*n* = 10 mice).

### 4.5. NaIO_3_-Induced Murine Model

Two-month-old 129 S2/SvHsd mice were subretinally injected in one eye with 4.5 × 10^8^ vg AAV2/8-VMD2-ophNdi1 and 1.0 × 10^8^ vg AAV2/2-CAG-EGFP [72], and in the other eye with 1.0 × 10^8^ AAV2/2-CAG-EGFP in PBS. At 5 months, mice were injected via tail vein with 22 mg/kg NaIO_3_ in 0.9% NaCl_2_. Mice underwent ERG analysis at 7 days and OKR analysis at 4 weeks post-insult (*n* = 7) [71,73]. OKR spatial frequency thresholds were measured blind on three occasions using a virtual optokinetic system (OptoMotry, CerebralMechanics, Lethbridge, AB, Canada). Treated and untreated eyes were compared using paired *t*-tests. Additionally, mice received 50 mg/kg NaIO_3_ in 0.9% NaCl_2_ via tail vein. Retinas were processed for histology 7 days post-insult as described (*n* = 7) [38].

### 4.6. Statistical Analysis

Statistical analysis was performed using GraphPad Prism (version 9.4, GraphPad Software, Boston, MA, USA). *t*-tests and ANOVA with post-hoc Tukey were considered significant at *p* < 0.05.

## Figures and Tables

**Figure 1 ijms-24-03847-f001:**
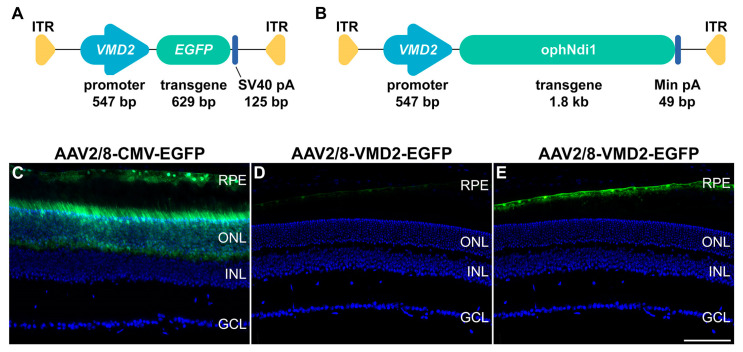
Diagrammatic representation of the plasmid constructs *VMD2*-ophNdi1 (**A**) and *VMD2*-EGFP (**B**), which were utilised to generate AAV2/8-VMD2-EGFP and AAV2/8-VMD2-ophNdi1 viral preparations. The *VMD2* promoter and *EGFP* and ophNdi1 transgenes, the minimal PolyA, ITRs and sizes are indicated. (**C**–**E**) Comparison of cell type specificity and EGFP expression level from the *VMD2* and *CMV* promoters in murine retina. Two-month-old 129 S2/SvHsd mice were subretinally injected with 3 × 10^9^ vg of AAV2/8-CMV-EGFP (**C**) or AAV2/8-VMD2-EGFP (**D**,**E**). Four weeks post-injection, eyes were enucleated, fixed in 4% PFA, cryosectioned, and processed for and analysed by fluorescent microscopy. Green and blue represent EGFP and DAPI (nuclear counterstain) fluorescence, respectively. Panels (**C**,**D**) were displayed using the full intensity range of EGFP fluorescence. Panel (**E**) corresponds to the same microscopy image as panel (**D**); however, in panel (**E**), the displayed EGFP fluorescence intensity range was focused on the lower intensity values, which enabled the visualisation of lower EGFP levels. EGFP fluorescence intensity levels were measured in the RPE for both promoters, and levels were found to be ~6.5-fold lower from *VMD2* compared to *CMV*. RPE: retinal pigment epithelium, ONL = outer nuclear layer, INL = inner nuclear layer, GCL = ganglion cell layer. Scale bar for retinal sections (**E**) = 100 μm.

**Figure 2 ijms-24-03847-f002:**
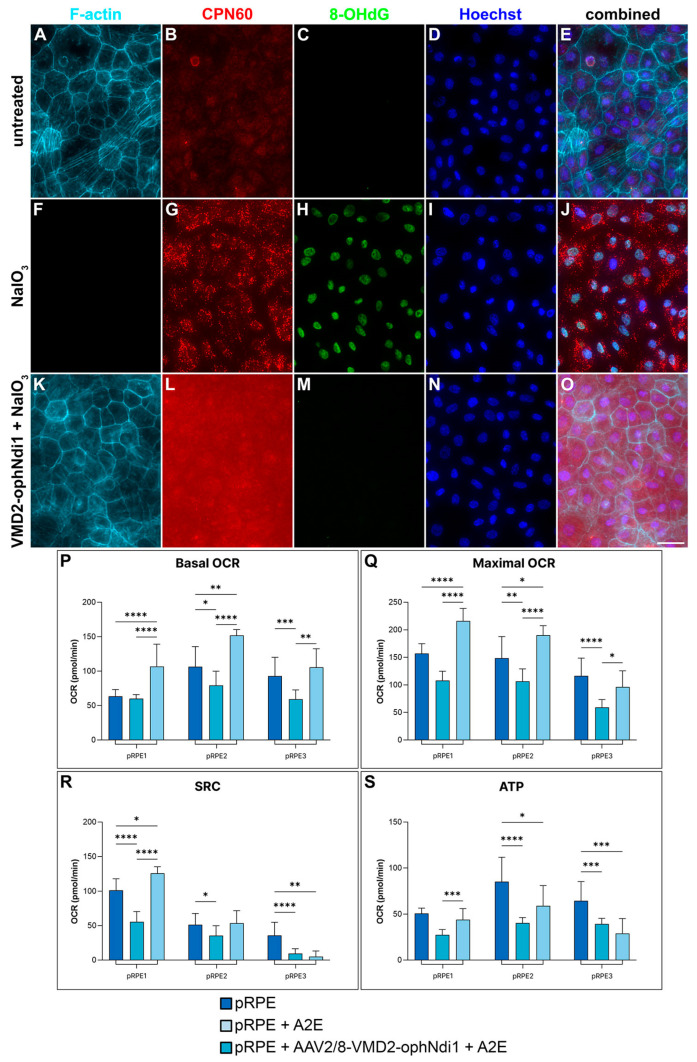
Rescue of primary pRPE cell models by AAV2/8-VMD2-ophNdi1. (**A**–**O**) Numbering 7.5 × 10^4^, pRPE cells (from *n* = 3 pigs) were transduced with AAV2/8-VMD2-ophNdi1 5 h post-seeding (MOI = 3.4 × 10^6^; (**K**–**O**)). Cells were insulted 28 h post-transduction with 6 mM NaIO_3_ (**F**–**O**) and 24 h post-insult cells were fixed and stained with Phalloidin-iFluor 647 (F-actin, light blue), CPN60 (mitochondrial marker, magenta) and 8-OHdG-Alexa Flour 488 (oxidative stress marker, green) immunocytochemistries; nuclei were counterstained with Hoechst (nuclear stain, dark blue). AAV2/8-ophNdi-treated and NaIO_3_-insulted cells (**K**–**O**) were compared to untreated control cells (**A**–**E**) and to untreated cells insulted with NaIO_3_ (**F**–**J**). (**P**–**S**) Numbering 5.0 × 10^4^, primary pRPE cells (*n* = 3 pigs; pRPE1-pRPE3) were seeded into XFe96 Seahorse plates. The following day, a minimum of five wells were transduced with AAV2/8-VMD2-ophNdi1 (MOI = 3.4 × 10^6^). Transduced cells and untransduced control cells (*n* > 15 wells) were insulted 28 h post-transduction with 30 μM A2E and exposed to 430 nm blue light of ~1 mW/cm^2^ for 3 h. (**P**) Basal and (**Q**) maximal oxygen consumption rates (OCRs), (**R**) spare respiratory capacity (SRC) and (**S**) ATP production were determined and are indicated. OCR was normalised to protein. Mitochondrial stress tests on pRPE1-3 were performed on 3 separate occasions * *p* < 0.05; ** *p* < 0.01; *** *p* < 0.001; **** *p* < 0.0001; ANOVA and post-hoc Tukey. Scale bar for immunocytochemistry panels (**O**) = 25 μm.

**Figure 3 ijms-24-03847-f003:**
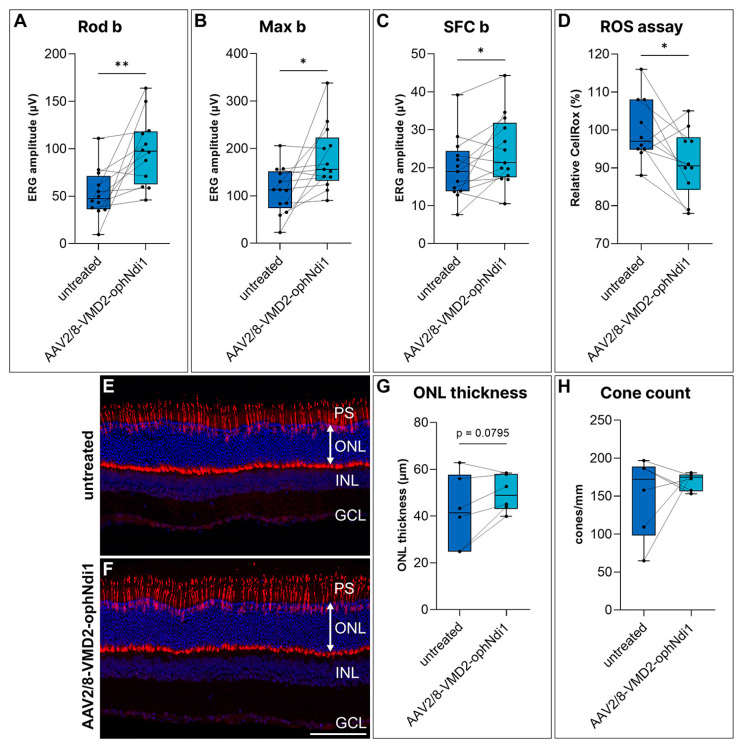
Rescue of the *Cfh*^−/−^ murine model with AAV2/8-VMD2-ophNdi1. Two-month-old *Cfh^−/−^* mice were subretinally injected with 4.5 × 10^8^ vg AAV2/8-VMD2-ophNdi1 in one eye, while contralateral eyes received PBS. (**A**–**C**) Electroretinography (ERGs) of treated eyes compared to untreated contralateral eyes at 8 months (*n* = 13). Rod b (**A**), Max b (**B**) and single flash cone (SFC) b (**C**) responses are indicated. (**D**) At 7 months post injections mice were sacrificed, retinal cells dissociated, stained with CellRox^TM^ (marker for ROS) and analysed on an Accuri B6 FACS analyser (*n* = 10 mice). Relative median levels of ROS in the single cell and live population of cells from treated and control retinas were compared. At 8 months eyes from untreated (**E**) and AAV2/8-VMD2-ophNdi1-treated (**F**) mice were fixed, cryosectioned and stained with Arr3 immunocytochemistry (cone-specific antibody, red) and DAPI (nuclear stain, dark blue). (**G**) The outer nuclear layer (ONL) thickness (μm) and (**H**) cone numbers (cones/mm) were determined (*n* = 6). PS = photoreceptor cell segments; INL = inner nuclear layer; GCL = ganglion cell layer. * *p* < 0.05; ** *p* < 0.01; paired *t*-test. Scale bar for retinal sections (**F**) = 100 μm.

**Figure 4 ijms-24-03847-f004:**
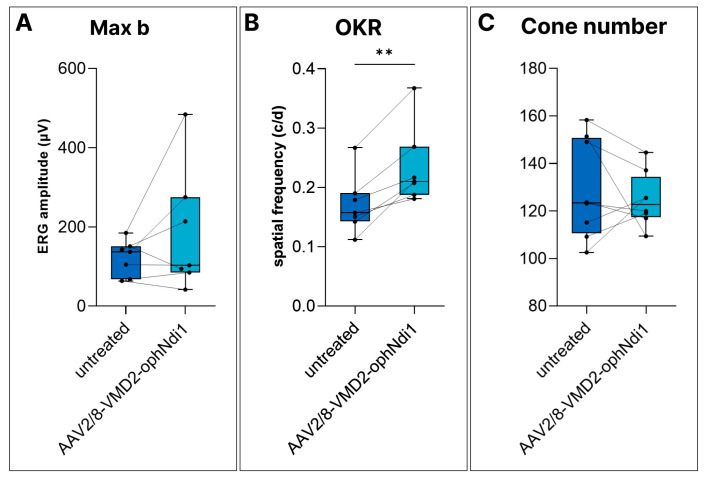
AAV2/8-VMD2-ophNdi1-mediated benefit in the NaIO_3_-induced murine model of AMD. (**A**,**B**) Adult 129 S2/SvHsd mice were subretinally injected with 4.5 × 10^8^ vg AAV2/8-VMD2-ophNdi1 and 1.0 × 10^8^ vg AAV2/2-CAG-EGFP in one eye, while contralateral eyes received an equal volume of PBS containing 1.0 × 10^8^ vg AAV2/2-CAG-EGFP (*n* = 7). At 3 months post-injection 22 mg/kg NaIO_3_ in 0.9% NaCl_2_ was administered via tail vein. (**A**) One week post-insult, ERG analysis was undertaken. (**B**) OKR analysis four weeks post-insult. (**C**) Additional adult 129 S2/SvHsd mice were injected with 4.5 × 10^8^ vg AAV2/8-VMD2-ophNdi1 and 1.0 × 10^8^ vg AAV2/2-CAG-EGFP in one eye, while contralateral eyes received an equal volume of PBS containing 1.0 × 10^8^ vg AAV2/2-CAG-EGFP (*n* = 7). Three months post-injection 50 mg/kg NaIO_3_ in 0.9% NaCl_2_ was administered via tail vein. Seven days post-insult mice were sacrificed, eyes fixed, retinas cryosectioned and stained with Arr3 immunocytochemistry (cone antibody, magenta) and DAPI (nuclear stain, dark blue). ** *p* < 0.01; paired *t*-test.

## Data Availability

The datasets used and/or analysed during the current study are available from the corresponding author on reasonable request.

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
