# Peer review of "RPE-Directed Gene Therapy Improves Mitochondrial Function in Murine Dry AMD Models"

_ijms, 2023, doi:10.3390/ijms24043847_

Round 1

Reviewer 1 Report

This manuscript by Millington-Ward et al. extends a previous study in which they identified a protective effect from AAV-vectored ophNdi1 under the control of a ubiquitous CMV promoter that directed expression of ophNdi1 to the RPE and photoreceptors from a subretinal injection. In this study, the authors ask the question of whether a more restrictive promoter (VMD2) could be used to target ophNdi1 expression to the RPE with similar efficacy. This study is significant since selective expression will benefit patients by restricting potential off-target expression.

In this study, the authors show that the AAV-VMD2-ophNdi1 is not as effective as AAV-CMV-ophNdi1 at preserving cone cells in Cfh-/- mice or in preserving ERG or cone cells in the NaI03 damage model. The authors suggest that this is because expression of ophNdi1 in both RPE and photoreceptors is required for the therapeutic benefit. However, there are several limitations to this study that prevent reaching this definitive conclusion. This principal problem lies in the dosing and expression analysis. The authors correctly identify that the VMD2 promoter drives weaker expression than the CMV promoter, and use a comparison analysis of GFP expression to determine that VMD2 is ~6.5-fold weaker than CMV. Accordingly, they use 6.5x as much AAV-VMD2-ophNdi1 in this study compared to their previous study. But one issue is that in the previous study a standardized dosing wasn’t used, instead delivering either 1.0 x 107 vg or 5.7 x 108 vg (see Fig 2 caption of Millington-Ward 2022). Depending on which dosing they choose, if they are targeting 6.5-fold higher dosing in this study, the dosing should be either 6.5 x 107 vg or 3.7 x 109 vg. Instead, they use 4.5 x 108 vg. So this dosing doesn’t match. However, this ‘dosing’ would be a moot point if some type of quantitative expression analysis was performed, preferably qRT-PCR or quantitative immunoblots or alternatively immunohistochemistry, to demonstrate that ophNdi1 expression in the RPE is actually being driven at a level similar to that desired. Without this piece of information it cannot be determined whether targeted expression of ophNdi1 to the RPE causes the loss of the protection or whether the VMD2 promoter is too weak to drive protective levels of expression.  Thus, this type of expression analysis must be added to the manuscript.

In addition, there are several minor points that should be addressed:

·        The supplemental figures should be included in the main text since they are essential to the paper

·        Figure S1 is never referenced in the text

·        For the study in RPE cells (Fig 1), a negative control for any AAV-induced effect should be included (e.g., an AAV that does not contain ophNdi1, such as AAV2/8-VMD2-GFP).

·        In Figure 1P-S, the biological replicates need to be combined; ie, RPE1-3 are each biological replicates and should be combined rather than treating them separately. Then the statistical analysis between the three conditions should be compared by ANOVA (rather than t-tests) with post-hoc comparison (matched pairing can be used for each of the RPE samples)

·        Histological images for the Cfh-/- should be shown in Figure 2 to enable independent assessment of the absence of cone rescue which was a significant benefit in the previous study; some assessment of rod histology should be performed (e.g., ONL thickness) since a rod benefit in ERG function was measured

·        The Figure 2 caption is repeated in the body of the text at the bottom of page 3 (Lines 137-148).

Author Response

Dear Reviewer,

Thank you for assessing our manuscript entitled, “RPE-directed therapy improves mitochondrial function in murine dry AMD models” for publication in IJMS (ijms-2037934). We thank you for your extremely valuable comments and have made alterations to the manuscript to address the points raised. We feel your input has improved to manuscript considerably.

Points raised:

The Reviewer notes that we have shown that the VMD2 promoter, which is specifically expressed in the RPE, is ~6.5-fold weaker than the ubiquitous CMV promoter at driving GFP expression in murine RPE. In a previous study we utilised two doses of CMV-ophNdi1 of 1 x 107 vg and 5.7 x 108 vg and found equivalent benefit in the Cfh-/- murine mouse, suggesting that the therapeutic dose of ophNdi1 is very wide (at least 57-fold). Based on this very wide dose range and that expression from the VMD2 promoter is ~6.5-fold lower than from the CMV promoter, we selected a dose of 4.5 x 108 vg per injection in the current study, which is a dose that should express within the therapeutic range as expression should fall neatly between the two doses of the previous study (1 x 107 vg and 5.7 x 108 vg).

The Reviewer suggests that it would be important for completion to confirm that expression of ophNdi1 from 4.5 x 108 vg VMD2-ophNdi1 is indeed similar in RPE to expression from 1 x 107 vg and 5.7 x 108 vg CMV-ophNdi1. The suggested study would need to be done in vivo as both transduction and expression of viral preparations, particularly from an RPE-specific promoter, in cell culture are unlikely to mimic the in vivo situation accurately.

However, there are a number of technical reasons why this is challenging. RT-qPCR and Western blot comparing expression profiles of ophNdi1 RNA or NDI1 protein in murine RPE would rely on one being able to extract RPE cleanly from murine retina, without contamination of photoreceptors and other cell types. However, most likely such contamination could lead to large errors, “in favour” of higher levels of observed expression from the ubiquitous CMV promoter, which expresses in all retinal cells.

Employing immunohistochemisty would rely on the availability of a specific antibody against NDI1, and this unfortunately is currently unavailable. We have tried a number of commercially available NDI1 antibodies and unsuccessfully tried to generate an antibody ourselves twice. The commercial antibodies, and our own ones only worked on Western blot and Elisa and not immunohistochemistry. Indeed this is the reason why in the current study AAV-CMV-EGFP and AAV-VMD2-EGFP were compared to give an indication of dose to use. Given the wide effective range of AAV-CMV-ophNdi1 from our previous study, it was decided that a dose in the middle of the two widely separated effective doses from the previous study should be used in the current study.

Minor points:

The supplemental figures have now been included in main text and are referenced correctly in the text, as suggested by the Reviewer.

In earlier studies AAVs expressing CAG-EGFP or rhodopsin promoter-driven rhodopsin (therefore not expressed in cells other than rod photoreceptor cells) have been assessed in a variety of cell types (hTERT-RPE1 and HEK293 cells) and have not been found to affect oxygen consumption rates. Hence in the current study a non-ophNdi1 AAV was not included as a control.

The Reviewer suggested that the OCR data from the primary RPE cells from three pigs should be combined and analysed together. The biological replicas were not combined because the baseline OCR reading from the cells were very distinct between different pigs. This is most likely due to inherent differences between pigs and also due to the fact that the RPE cells are primary and vary due to small differences in culturing (e.g. exact age of the cells, and at what stage in replication they were analysed). These variations cannot be controlled. However, primary cells from different patients or different animals are often kept separate for this reason or, indeed, frequently only data from one patient or animal are provided (for example; Fu et al., Neuro Oncol 22(2) 2020; Buettner at al., J Hematol and Oncol 14 2021; Chadderton et al., Pharmaceutics 2023). The Reviewer also suggested that the different treatment groups (cells without treatment, insulted cells and insulted cells treated with AAV) should be analysed using ANOVA instead of paired t-tests. We have taken this helpful suggestion on board and re-analysed the statistics accordingly (Figure 2P-S).

The Reviewer asked for histological images of the stained Cfh-/- mouse retinas to be included and rod photoreceptor cell histology analysed. Histological images for the Cfh-/- mouse retinas stained with Arr3 (cone stain) and dapi (nuclear stain) have now been included in Figure 3E and F. In addition, the ONL thickness was measured and is included in a separate chart in Figure 3G.

The errant caption has been removed from the bottom of page 3.

The Reviewers commented that extensive revisions of the English should be made. This was surprising to us in Dublin, as we are native English speakers. However, we did find quite a number of typographical errors, which we apologise for and which we have amended. If these corrections are not sufficient, respectfully we wonder would the Reviewer please indicate the region(s) of the manuscript that are of concern.

Reviewer 2 Report

The “RPE-directed gene therapy provides benefit mitochondrial dys-2 function in murine dry AMD models” article shows the results of using gene therapy with a restrictive RPE promoter to localize its action only in those cells for therapy in porcine and murine models of Dry-AMD. it is a well-designed study that shows, thanks in part to previous studies with general promoters, some very interesting results on the need to act on both the RPE and photoreceptors.

However, there are certain errors that should be corrected in order to be published:

-in row 132 figure S1 is mentioned but it should be Figure S2.

-From row 137 to row 148 the footnote of figure S2 is exposed as if they were results, the authors should eliminate that paragraph.

-In material and methods, it is not indicated if the study with animals has been approved by an ethical committee for animals or if they comply with the regulations for the use of animals in their country. This information needs to be added.

-Regarding the statistical analysis, the comparisons between the groups in Figures 1P-S should have been made using Anova and a posteriori test and not using Student's T.

-I think there is an excess of bibliography, it should be limited to the most relevant and current

Author Response

Dear Reviewer,

Thank you for assessing our manuscript entitled, “RPE-directed therapy improves mitochondrial function in murine dry AMD models” for publication in IJMS (ijms-2037934). We thank you for your extremely valuable comments and have made alterations to the manuscript to address the points raised. We feel your input has improved to manuscript considerably.

Points raised:

The Reviewer has requested that supplemental Figures S1 and S2 be included in the main text. These have therefore been combined into a new Figure 1 in the main text and are referred to appropriately in the text.

The errant caption has been removed from the bottom of page 3.

The material and methods have been amended to indicate that the study with animals was approved by our ethical committee for animals and complies with the regulations for the use of animals in Ireland.

The Reviewer also suggested that the different treatment groups (cells without treatment, insulted cells and insulted cells treated with AAV) should be analysed using ANOVA instead of paired t-tests. We have taken this helpful suggestion on board and re-done the statistics accordingly (Figure 2P-S).

The Reviewer was concerned that the bibliography contained too many non-relevant references. I have removed 7 less relevant references from the manuscript.

Round 2

Reviewer 1 Report

The authors have adequately addressed the previous concerns. One note on line 222, the figure reference should be Figure 3D (not 3E).

Author Response

Dear Reviewer,

Many thanks for doing such a throrough job reviewing the manuscript and even spotting that misreferenced figure in line 222. I have corrected it to read 3D.

Kind regards,

Sophia